# The Non-Invasive Ultrasound-Based Assessment of Liver Viscosity in a Healthy Cohort

**DOI:** 10.3390/diagnostics12061451

**Published:** 2022-06-13

**Authors:** Alexandru Popa, Ioan Sporea, Felix Bende, Alina Popescu, Renata Fofiu, Andreea Borlea, Victor Bâldea, Ariana Pascu, Camelia Gianina Foncea, Radu Cotrău, Roxana Șirli

**Affiliations:** 1Division of Gastroenterology and Hepatology, Department of Internal Medicine II, Center for Advanced Research in Gastroenterology and Hepatology, “Victor Babes” University of Medicine and Pharmacy, E. Murgu Square, Nr. 2, 300041 Timisoara, Romania; popa.alexandru@umft.ro (A.P.); isporea@umft.ro (I.S.); bendefelix@gmail.com (F.B.); alinamircea.popescu@gmail.com (A.P.); victorbaldea07@gmail.com (V.B.); pascuariana@yahoo.com (A.P.); foncea.camelia@gmail.com (C.G.F.); cotrau.radu@yahoo.com (R.C.); roxanasirli@gmail.com (R.Ș.); 2Division of Endocrinology, Department of Internal Medicine II, “Victor Babes” University of Medicine and Pharmacy, E. Murgu Square, Nr. 2, 300041 Timisoara, Romania; andreea.brl3@gmail.com

**Keywords:** liver fibrosis, two-dimensional shear-wave elastography, liver inflammation, viscosity, healthy subjects

## Abstract

Liver fibrosis is the most significant prognostic factor in chronic liver disease (CLD). Clinical practice guidelines recommend the use of non-invasive techniques, such as two-dimensional shear-wave elastography (2D-SWE), to assess liver stiffness as a marker of fibrosis. Several other factors influence liver stiffness in addition to liver fibrosis. It is presumed that changes due to necro-inflammation modify the propagation of shear waves (dispersion). Therefore, new imaging techniques that investigate the dispersion properties of shear waves have been developed, which can serve as an indirect method of measuring liver viscosity (Vi PLUS). Defining the reference values in healthy subjects among different age groups and genders and analyzing the factors that influence these values is essential. However, published data on liver viscosity are still limited. This is the first study that aimed to assess the normal range of liver viscosity values in subjects with healthy livers and analyze the factors that influence them. One hundred and thirty-one consecutive subjects with healthy livers were enrolled in this prospective study. The results showed that Vi PLUS is a highly feasible method. Liver stiffness, age and BMI influenced the liver viscosity values. The mean liver viscosity by Vi PLUS in subjects with healthy livers was 1.59 Pa·s.

## 1. Introduction

An estimated 2 million deaths each year are attributed to chronic liver diseases (CLD), making it a global public health problem [1]. Considering the high rates of treatment response in patients infected with hepatitis C virus (HCV) and improved management of patients infected with hepatitis B virus (HBV) treated with analogues, hepatologists have switched their focus to non-alcoholic fatty liver disease (NAFLD) and alcohol-related liver disease (ALD), as their incidence in developed countries is increasing. Regardless of etiology, liver injury causes a series of inflammatory events that lead to chronic inflammation. Chronic inflammation is a dynamic process that induces the development of liver fibrosis (LF), eventually prompting the progression to cirrhosis and hepatocellular carcinoma. As it marks a turning point in the evolution of CLD, LF is considered one of the most significant prognostic factors [2]. Although liver biopsy is the gold standard for diagnosing and grading LF, it has significant drawbacks: it is invasive and is prone to sampling errors in addition to posing the risk of significant complications [3]. Therefore, there has been a shift toward non-invasive techniques (NITs). Currently, the two main types of NITs to assess LF are serological biomarkers and imaging-based elastography methods. The imaging techniques include transient elastography (TE), shear-wave elastography (SWE) and magnetic resonance elastography (MRE) [4,5,6].

The newest of the SWE techniques is two-dimensional SWE (2D-SWE), which is an ultrasound-based method that allows a real-time tissue examination. Liver stiffness (LS) as a marker of fibrosis is assessed by measuring the propagation speed of shear waves generated into the tissue by ultrasound (US) pulses. Elasticity is illustrated as a color-coded elastogram, enabling both a qualitative and quantitative LS estimation. Several studies ascertained the usefulness of 2D-SWE in assessing the stiffness of different organs such as the liver, spleen, or thyroid [5,7,8,9,10].

LS is influenced by several other factors in addition to LF. Necro-inflammation, cholestasis, liver venous congestion, and food ingestion may influence LS, consequently acting as confounding factors [11,12,13]. Steatosis, cholestasis, and venous congestion are easily determined by standard ultrasound, but necro-inflammation remains the only one of these factors assessable only by liver biopsy.

Ideally, the hepatic shear-wave speed would increase uniformly with increasing fibrosis in a simple and precise manner not influenced by other factors. However, biological soft tissues are rather viscoelastic, and LS is influenced by both elasticity and viscosity. In rheological models of viscoelastic material, viscosity (Pa·s) is represented as a damper, and elasticity is represented as a spring (kPa). Viscosity is the measure of resistance to shearing motion. The tissue shows movement under gradual deformation instead of sudden deformation. It is presumed that changes due to necro-inflammation modify the shear-wave propagation (viscosity) [14]. However, the current algorithms used for assessing LS neglect the viscosity properties of different tissues. Several studies showed that LS increases significantly in early non-alcoholic steatohepatitis or alcoholic liver disease even if fibrosis has not yet developed; therefore, an overestimation of LF can appear [12,15]. Thus, a more accurate assessment of liver pathology would be attained by evaluating both the elasticity and the viscosity [14].

US manufacturers have recently developed new imaging techniques that investigate the dispersion properties of shear waves that can serve as an indirect method for measuring viscosity. The Hologic SuperSonic Mach 30 equipped with the new UltraFast software enables the simultaneous real-time quantification of LS by shear-wave elastography (2D-SWE) and of liver viscosity by Viscosity Plane Wave UltraSound (Vi PLUS). However, published data on liver viscosity are still limited. Therefore, establishing the normal liver viscosity values and their normal variability in healthy subjects is a necessity in order to differentiate normal from pathological [16,17]. To our knowledge, this is the first study that aims to assess the normal ranges of liver viscosity values and to analyze the effects of gender, age, and body mass index (BMI) in a large cohort of subjects with healthy livers.

## 2. Materials and Methods

### 2.1. Study Population

In this prospective monocentric study conducted between October 2019 and October 2021 in a tertiary Gastroenterology and Hepatology center, 131 consecutive subjects with healthy livers were enrolled.

All subjects agreed to undergo elastographic measurements as well as clinical, ultrasound and biological examinations and provided written consent before study entry. The study was approved by the local research ethics committee and the review board of our university (042/10 December 2018). It was performed in accordance with the last revised version of the World Medical Association Declaration of Helsinki (revised in 2000, Edinburgh). 

The inclusion criteria were: age older than 18 years, normal abdominal US examination, normal LS values evaluated by TE (LS < 6 kPa), normal Controlled Attenuation Parameter (CAP) value (lower than 248 dB/m) [18,19], without obesity (BMI lower than 30) [20], normal blood count, normal liver function tests (ALT (alanine transaminase): 0–55 U/L, AST(aspartate transaminase): 5–34 U/L, albumin: 3.4–5 g/dL, total bilirubin: 0.2–1.2 mg/dL), normal INR (International Normalized Ratio): 0.8–1.07, and negative HBV/HCV infection (negative hepatitis B antigen/negative anti-hepatitis C virus antibody), no oncological history, no history of chronic hepatopathy, no history of alcohol abuse (defined as ≥3 drinks/day for men, ≥2 for women) [21], and no cardiovascular disease and/or heart failure [22].

Exclusion criteria were: pregnant women and subjects who refused to provide informed consent.

Data collected included age, gender, BMI, abdominal circumference, complete blood counts, international normalized ratio, total bilirubin concentrations, and aminotransferases levels.

### 2.2. Examination Protocol

The subjects fasted for a minimum of 4 h before examinations. Firstly, TE and CAP measurements were performed with a FibroScan^®^ Compact 530 device (EchoSens^®^, Paris, France). If the TE measurements were below 6 kPa, the CAP values were below 248 dB/m and the participants met the other inclusion criteria, the subjects were furtherly examined using the Supersonic MACH^®^ 30 US system (Hologic^®^ SuperSonic^®^ Imagine, Aix-en-Provence, France). Gray-scale US, as well as 2D-SWE and Vi PLUS measurements, were performed. All measurements were made by physicians with at least three years’ experience in US and US-based elastography, blinded to the patient’s clinical data. Subjects were examined in the supine position with the right arm in maximal abduction after they had rested for at least 10 min. The probe was placed between the ribs parallel to the intercostal space.

### 2.3. Transient Elastography and Controlled Attenuation Parameter

TE and CAP measurements were performed in all subjects using FibroScan^®^ Compact 530, using the standard M (3.5 MHz frequency) probe or the XL (2.5 MHz frequency) probe. The automatic probe selection tool was used to choose the appropriate probe. Reliable results (the median value of 10 valid measurements) were considered those with an interquartile range interval (IQR) to the median ratio (IQR/M) < 30% [23]. The measurements were expressed in kilopascals (kPa) with values ranging between 2.5 and 75 kPa for liver stiffness and between 100 and 400 dB/m for steatosis. 

### 2.4. Shear-Wave Elastography

A C6-1X single-crystal curved transducer (1 MHz to 6 MHz frequency) was used to perform the 2D-SWE measurements. The 2D-SWE mode displays tissue elasticity in the form of an easy-to-interpret color-coded image (Figure 1) and quantitative data. Local estimation of tissue stiffness is expressed in kPa or m/s over a wide range of values. The 2D-SWE measurement box was placed at least 1 cm below the liver capsule in an area free of other structures. Once the 2D-SWE map was appropriate, the patient was asked to hold his breath while an image acquisition was performed. Then, the Q-Box™ was placed over an area of relative homogeneous elasticity at a 3–5 cm depth. A reliable result was defined as the median value of five 2D-SWE measurements (obtained from five frames at a stability index SI > 90%) with an IQR/M < 30%.

### 2.5. Viscosity PLUS

Since Vi PLUS (Figure 1) is an additional parameter obtained at the same time as the 2D-SWE measurement, the same acquisition protocol and same US probe as in 2D-SWE was used. Vi PLUS analyzes the propagation speed of the shear wave at several frequencies and delivers information regarding the tissue shear-wave dispersion. The variations of the shear waves’ velocity among frequencies are illustrated in the form of a color-coded map as well as a numerical value, which is expressed in pascal-second (Pa·s) over a range of values from 1.0 to 5.0 Pa·s.

### 2.6. Statistical Analysis

The statistical analysis was performed using MedCalc Version 19.4 (MedCalc Software Ltd., Ostend, Belgium) and Microsoft Office Excel 2019 (Microsoft^®^, Redmond, Washington, DC, USA). Descriptive statistics were applied for demographic, anthropometric and laboratory findings. The Kolmogorov–Smirnov test was used for testing the distribution of numerical variables. Continuous variables were presented as mean and standard deviation (SD), while categorical variables were presented as frequency and percentages. Group comparisons of categorical variables were performed using Pearson’s x^2^-test. 

The individual impact of several parameters on the variance of Vi PLUS measurements was assessed by using linear regression analysis and multivariate regression models. The predictors, in the final regression equations, were accepted according to a repeated backward-stepwise algorithm (inclusion criteria *p* < 0.05, exclusion criteria *p* > 0.10) in order to obtain the most appropriate prediction model. Then, 95% confidence intervals (CI) were calculated for each predictive test, and a *p*-value below 0.05 was considered to concede statistical significance.

## 3. Results

### 3.1. Baseline Characteristics

One hundred and thirty-one consecutive adult subjects without known liver pathology who underwent multiparametric US-based measurements were enrolled. Of these, 8/131 (6.1%) patients had invalid or unreliable US-based measurements; 123 subjects were included in the final analysis. The baseline characteristics, demographic data, laboratory parameters, and LS values of the patients with reliable measurements are presented in Table 1.

### 3.2. Feasibility of 2D-SWE and Vi PLUS

Using 2D-SWE and Vi PLUS, valid measurements were obtained in 93.9% (123/131). Failure to acquire valid measurements with 2D-SWE and Vi PLUS in 2/8 patients was due to an inhomogeneous filling of the color map (no or little signal). The rest of the unreliable measurements were considered as such because of IQR/M > 30% or of the SI < 90%. 

Abdominal circumference mean values were significantly higher for patients with unreliable measurements as compared to those with reliable measurements (95.75 ± 6.26 cm vs. 84.70 ± 12.06 cm, *p* = 0.0115), while no significant differences were found for BMI mean values (25.35 ± 2.73 kg/m^2^ vs. 24.02 ± 3.48 kg/m^2^, *p* = 0.2410).

The feasibility, defined as the likelihood of obtaining a valid measurement, was analyzed. The combination of 2D-SWE and Vi PLUS had a very good feasibility of 93.9%.

### 3.3. Vi PLUS Values in Subjects with Healthy Livers and the Influence of Subjects’ Characteristics on Vi PLUS

The mean liver Vi PLUS value obtained in subjects with healthy livers (n = 123) was 1.57 ± 0.20 Pa·s for females and 1.62 ± 0.21 Pa·s for males, respectively. No significant differences between Vi PLUS mean values were found (*p* = 0.1872) (Figure 2). 

The distribution of Vi PLUS values is illustrated in Figure 3.

Vi PLUS mean values according to age subgroups are summarized in Table 2. 

Vi PLUS mean values increased with each decade of age (Figure 4). Mean values were significantly lower in the 18–30 years subgroup compared to all the other subgroups (all *p* < 0.05). In addition, significant differences were found between the 31–40 years group and the 51–60 years group (*p* = 0.0276). No differences were found between Vi PLUS mean values for subjects in the 41–50 years subgroup compared to those from the 51–60 and 61–80 years subgroups (*p* = 0.1115 and *p* = 0.1759, respectively), nor between subjects from the 51–60 years subgroup and those from the 61–80 years one (*p* = 0.8102). 

According to BMI (kg/m^2^), 73/123 (59.4%) were normal weight subjects (BMI (kg/m^2^) < 25), while 50 /123 (40.6%) were overweight subjects (25 ≤ BMI (kg/m^2^) < 30). Mean Vi PLUS values were significantly lower in subjects with normal weight (1.53 ± 0.19 Pa·s) compared to overweight subjects (1.67 ± 0.19 Pa·s) (*p* = 0.0001) (Figure 5). 

In univariate regression analysis, the following parameters were associated with Vi PLUS values: age (*p* < 0.001), BMI (*p* < 0.001), abdominal circumference (*p* < 0.001), LS values by FS (*p* < 0.001) and LS values by 2D-SWE (*p* < 0.001), respectively. Multivariate regression analysis was used to evaluate the independent factors associated with Vi PLUS values. The regression model was built based on the forward stepwise method, and Akaike information criteria (AIC) were used to appreciate the best model. The model including age (*p* = 0.0043), BMI (*p* = 0.0023), and LS values by 2D-SWE (*p* < 0.0001) was associated with Vi PLUS values. In addition, a good correlation between Vi PLUS measurements and LSM by 2D-SWE (r = 0.66, 95%CI: 0.55–0.75, *p* < 0.0001) was found in normal subjects. 

## 4. Discussion

The stage of fibrosis is known to be a factor associated with mortality in CLD. Although liver biopsy is the gold standard for diagnosing and grading LF, it has important drawbacks [3]. Therefore, there has been a shift toward non-invasive techniques (NITs) [24]. A substantial number of studies using LB as a reference have shown good accuracy of the old 2D-SWE technique developed by SuperSonic Imagine to predict different stages of fibrosis in CLD [25,26,27]. To date, many clinical guidelines recommend using non-invasive tests for the detection and staging of liver fibrosis [5].

Ideally, the hepatic shear-wave speed would increase uniformly with increasing fibrosis in a simple and precise manner that is not influenced by other factors. However, biological soft tissues are rather viscoelastic than entirely elastic [28]. Therefore, the role of cofactors can be major. In addition to liver fibrosis, several factors, such as the presence of steatosis or necro-inflammation, influence the viscoelastic properties of the liver tissue. Given that these pathological conditions often coexist, it is crucial to determine if the LS is increased due to inflammation or fibrosis. Therefore, US producers have developed new parameters which are aiming to better assess CLD patients. Such a parameter is liver viscosity, which is an imaging technology based on the shear-wave dispersion, and it is considered to be a surrogate of necroinflammation [11,16,29].

Several clinical studies have sought to evaluate the role of liver viscosity in the assessment of CLD. Deffieux et al. were the first to publish data evaluating liver viscosity using the SuperSonic US device. The clinical prospective study, which included 120 patients with various CLD, showed that viscosity was less efficient in staging liver fibrosis than 2D-SWE and also was a modest predictor of disease activity [30]. Chen et al. evaluated both stiffness and viscosity in a study that included 45 patients. The results showed that viscosity is less efficient in evaluating liver fibrosis compared with 2D-SWE [29]. Sugimoto et al. revealed that stiffness was more useful than viscosity for predicting the stage of fibrosis. However, in contrast with Deffieux’s study, viscosity was found to be useful for predicting the degree of necroinflammation [16,31]. Moreover, a recently published study found higher Vi PLUS liver values in COVID-19 patients with pulmonary injury compared to COVID-19 patients without pulmonary injury [32].

When new techniques are introduced on the market, the feasibility analysis is essential to establish their clinical applicability. Two-dimensional elastography techniques that are already in use have shown to be very useful for liver fibrosis assessment. Several studies have revealed that reliable LSMs can be obtained in 90–98% of patients [33,34,35]. This study showed that the simultaneous measuring of liver viscosity and stiffness using the new software embedded into SuperSonic Mach 30 is highly feasible, with 93.9% feasibility. A clinical prospective study, which included 120 patients, published by Deffieux et al. showed that the viscosity measurement technique from SSI had a 97.5% feasibility [30].

To properly understand these new US-based parameters, defining the reference values in subjects with healthy livers among different age groups and genders and analyzing the factors that influence these values is essential. Nevertheless, to our knowledge, no other clinical study focused on defining the reference values of liver viscosity in subjects with healthy livers. In our study, the mean viscosity value in subjects with healthy livers was 1.59 ± 0.36 Pa·s. Consequently, a Vi PLUS measurement of around 1.59 Pa·s can be considered as indicative of a normal liver without liver fibrosis or inflammation. Vi PLUS mean values increased with age but were not influenced by gender. A study published by Sabira et al. in 2021 showed that an age-associated increase in necroptosis contributes to chronic inflammation in aging liver “inflammaging” [36,37]. Inflammaging refers to the chronic, low-grade macrophage-centered inflammation of different tissues including the liver that characterizes aging [38]. 

In order to reduce the confounding by hepatic steatosis or fibrosis on viscosity values, subjects with fatty liver were excluded using B-mode ultrasound and CAP; also, subjects with significant fibrosis were excluded using TE. However, the mean Vi PLUS values obtained in subjects with normal weight were significantly lower compared to those obtained in overweight subjects. A studied published by Luo et al. in 2021 concluded that obesity is accompanied by a high level of inflammatory factors that can lead to steatosis development. Another article published by Casagrande et al. in 2020 showed that hepatic inflammation preceded hepatic steatosis [39]. Therefore, changes in shear-wave dispersion due to liver inflammation could precede the changes due to steatosis development. The univariate regression analysis shows that LS has a strong and significant effect on viscosity values (r = 0.73, *p* < 0.001). As mentioned at the beginning of the discussions section, several clinical reports published uncovered a similar effect [16,30,31].

However, further studies on patients with chronic hepatopathies, using liver biopsy as a reference method, are needed for a better understanding of the various factors that influence liver viscosity. The lack of LB in the present study is one of the main limitations. Still, the present study included volunteers with no previous liver disease, and obtaining a liver biopsy in this category of patients is challenging [5,6,7]. 

In addition to these limitations, the study is the first study to analyze the values of hepatic viscosity in a large cohort of subjects with healthy livers and provides essential information in this regard.

## 5. Conclusions

Vi PLUS by SSI is a highly feasible method. Liver stiffness, age and BMI influenced the liver viscosity values. The mean liver viscosity determined by Vi PLUS was significantly higher in overweight individuals than in normal-weight subjects (1.67 vs. 1.53 Pa·s). Vi PLUS values increased with age. The overall mean value of liver viscosity in the cohort of participants with healthy livers was 1.59 Pa·s. 

## Figures and Tables

**Figure 1 diagnostics-12-01451-f001:**
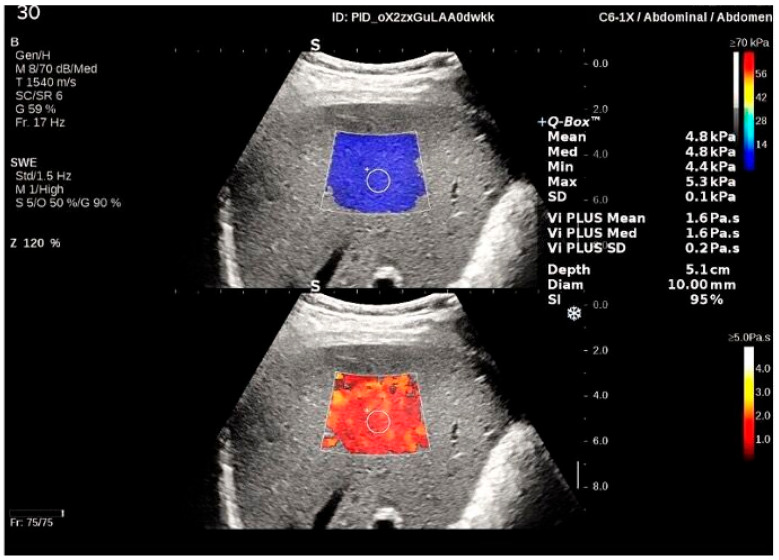
Illustration of a two-dimensional shear-wave elastography (2D-SWE) and a viscosity (Vi PLUS) measurement made in a healthy subject. Two-color scale maps are shown. In the upper half of the image, the 2D SWE map is displayed. Low stiffness is color-coded with blue, while red signifies high stiffness. The viscosity map is illustrated in the lower part of the image. Colors close to red indicate low viscosity, while yellow–white represents high viscosity. Quantitative results of 2D-SWE (expressed in kPa) and Vi PLUS (expressed in Pa·s) are displayed in the right part of the image. The mean, median, minimum, maximum, and standard deviation (SD) of the measurements, along with the depth, the diameter of the region of interest (ROI), and the Stability Index (SI) are also presented.

**Figure 2 diagnostics-12-01451-f002:**
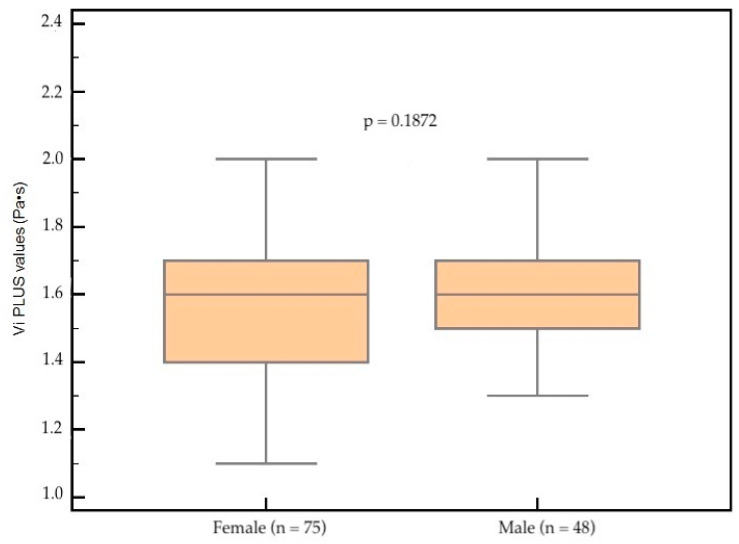
Vi PLUS values according to gender. No significant differences between Vi PLUS mean values were found (*p* = 0.1872).

**Figure 3 diagnostics-12-01451-f003:**
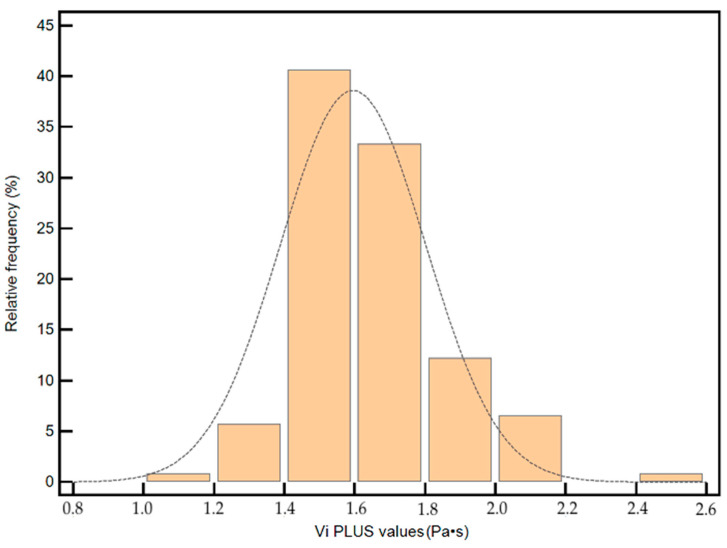
The distribution of Vi PLUS values in normal subjects. Approximately 88% of Vi PLUS values were in the range (1.4–1.8).

**Figure 4 diagnostics-12-01451-f004:**
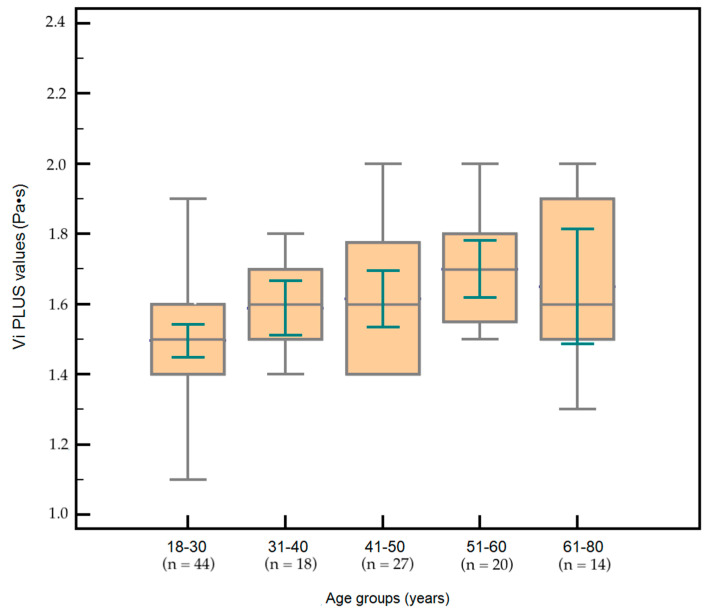
Box-and-whisker distribution plots comparing Vi PLUS values according to five different age subgroups. VI PLUS values slightly increased with each decade of age.

**Figure 5 diagnostics-12-01451-f005:**
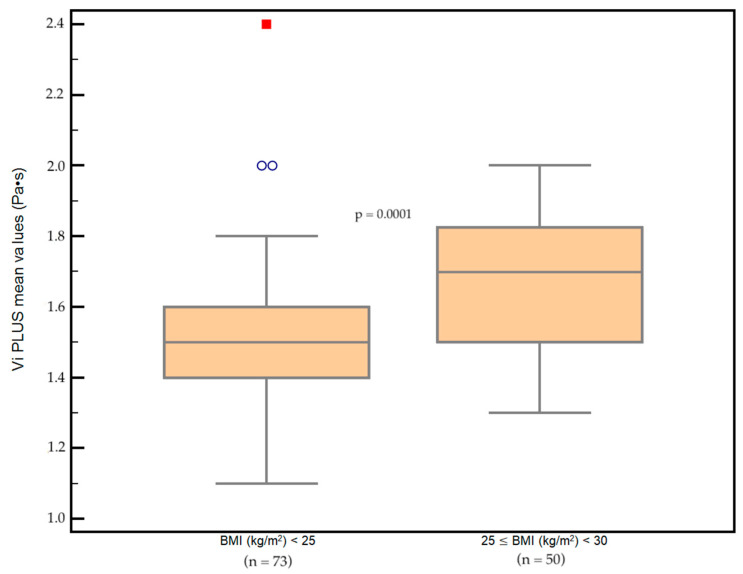
Box-and-whisker distribution plots comparing Vi PLUS values according to BMI. Vi PLUS values were significantly higher in overweight subjects (*p* = 0.0001). The two small blue circles and the red square are the graphical representation for outliers (an extremely high or extremely low value).

**Table 1 diagnostics-12-01451-t001:** Characteristics of subjects with reliable measurements.

Parameter	Normal Subjects n = 123
Mean age (years)	41.23 ± 13.40
Gender	
Males	48/123 (39%)
Females	75/123 (61%)
Mean BMI (kg/m^2^)	24.02 ± 3.48
Abdominal circumference (cm)	84.70 ± 12.06
AST (UI/L)	29.99 ± 12.90
ALT(UI/L)	32.17 ± 16.60
GGT (mg/dL)	63.21 ± 42.57
Cholesterol (mg/dL)	190.16 ± 39.93
Triglyceride (mg/dL)	135.96 ± 43.29
Platelet count (×10^9^ /L)	245.20 ± 68.32
LS by TE (kPa)	4.24 ± 1.18
2D-SWE (kPa)	4.98 ± 0.99
Vi PLUS (Pa·s)	1.59 ± 0.20
CAP (dB/m)	179.99 ± 51.65

Data are presented as numbers and percentages or mean ± standard deviation. ALT = alanine aminotransferase, AST = aspartate aminotransferase, BMI = body mass index, GGT = gamma-glutamyl transferase, LS = liver stiffness, TE = transient elastography, 2D-SWE = two-dimensional shear-wave elastography by SuperSonic Imagine, Vi PLUS = viscosity plane wave ultrasound.

**Table 2 diagnostics-12-01451-t002:** Vi PLUS mean values according to age subgroups.

Age Subgroups	Vi PLUS Mean Values (Pa·s)
18–30 years: 44/123	1.49 ± 0.14
31–40 years: 18/123	1.58 ± 0.15
41–50 years: 27/123	1.61 ± 0.20
51–60 years: 20/123	1.70 ± 0.17
61–80 years: 14/123	1.72 ± 0.31

Data are presented as number or mean ± standard deviation; Vi PLUS = Viscosity Plane Wave Ultrasound.

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
