# Peer review of "The Non-Invasive Ultrasound-Based Assessment of Liver Viscosity in a Healthy Cohort"

_diagnostics, 2022, doi:10.3390/diagnostics12061451_

Round 1
Reviewer 1 Report
The authors recruited 199 so-called “healthy adults” to explore the liver viscosity value in “healthy” liver. There were 178 subjects finally for analysis and a value of 1.69 Pa.s was concluded.
Some comments were raised as follows.
- Can the authors define the “viscosity” in liver? I believe most readers are interested in liver viscosity and the definition of “liver viscosity” will help the readers get more understanding.
- The authors should define the word “healthy” in their 199 healthy adults. There were 98 subjects with BMI >25 kg/m2, which is the cut-off value of over-weight or obesity. Were these subjects really “healthy”?
- The values of liver viscosity were significantly different among age groups and BMI groups, it was inadequate to conclude a single value of viscosity for healthy liver.
- Without liver pathology, how did the authors confirm whether there was no necro-inflammation in liver even under normal ALT in “healthy adults”?
- Since liver viscosity was less efficient in staging liver fibrosis reported in past studies and viscosity was only useful for predicting the degree of necroinflammation, what is the clinical importance of viscosity value in clinical practice? Liver fibrosis is an important prognostic factor in chronic liver disease, but not necroinflammation.
- The authors exclude subjects with fatty liver using B-mode ultrasound. Were the subjects with BMI >25 really without any fatty liver? CAP could be got by Fibroscan Compact 530 device. Why did the authors not use CAP to evaluate steatosis?
- Metabolic syndrome or DM, hypertension was supposed to be considered in the exclusion criteria.
- The relationship between steatosis and liver viscosity is unknown. The authors should address this issue.
- The normal references of AST, ALT, GGT, cholesterol, triglyceride and platelet should be listed in the manuscript.
Author Response
We want to thank the reviewers for the kind words of appreciation. It means a lot to us. Also, we want to thank you for the suggestions that have certainly helped us to improve this article significantly. We have made significant changes and hope that the publication of this article will encourage further research on this topic.
Reviewer 1
Some comments were raised as follows.
- Can the authors define the “viscosity” in liver? I believe most readers are interested in liver viscosity and the definition of “liver viscosity” will help the readers get more understanding.
Thank you, changes have been made to define liver viscosity better and to better define its importance. Introduction, page 2, lines 59-70.
- The authors should define the word “healthy” in their 199 healthy adults. There were 98 subjects with BMI >25 kg/m2, which is the cut-off value of over-weight or obesity. Were these subjects really “healthy”?
Thank you very much for this important remark. Since obesity was officially recognized as a chronic disease back in 2013, we decided to correct this mistake and exclude all the obese subjects from this study (defined as BMI> 30). Moreover, the lot was reanalyzed, taking into consideration values obtained using the controlled attenuation parameter(CAP), fibroscan. We included in the final analysis only subjects that had CAP values below the cut-off of 248 dB/m(no steatosis) as defined by Karlas. The final number of subjects included dropped to 123, but we consider that we have avoided many biases doing so. Thus, significant changes were made to the entire manuscript. Material and methods/Results/Conclusions
- The values of liver viscosity were significantly different among age groups and BMI groups, it was inadequate to conclude a single value of viscosity for healthy liver.
Thank you, we reformulated the conclusions so that they better reflect the results of the study.
- Without liver pathology, how did the authors confirm whether there was no necro-inflammation in liver even under normal ALT in “healthy adults”?
The certainty of the absence of necroinflammation could only be made based on liver biopsy, as we have mentioned in the limitations section. But we excluded liver pathology by using abdominal ultrasound, biological tests, liver fibrosis quantification by using TE and SWE, liver steatosis quantification by using standard gray scale ultrasound and CAP. On the other hand, to date, none of the studies published used liver biopsy when assessing values in healthy individuals for reasons that are easy to understand.
Ex:
https://pubmed.ncbi.nlm.nih.gov/25282481/
https://www.cghjournal.org/article/S1542-3565(18)30948-0/fulltext
https://journals.plos.org/plosone/article?id=10.1371/journal.pone.0203486
https://pubmed.ncbi.nlm.nih.gov/29249457/
- Since liver viscosity was less efficient in staging liver fibrosis reported in past studies and viscosity was only useful for predicting the degree of necroinflammation, what is the clinical importance of viscosity value in clinical practice? Liver fibrosis is an important prognostic factor in chronic liver disease, but not necroinflammation.
Liver fibrosis is the most important prognostic factor in CLD, but liver stiffness assessed by elastography (the method recommended by guidelines as a NIT for evaluating liver fibrosis) is not only affected by the presence of fibrosis (liver is viscoelastic rather than elastic alone). Several other confounding factors, such as food intake, the presence of hepatic steatosis, congestion, cholestasis, or inflammation, can increase stiffness even in the absence of fibrosis. If food intake, the presence of hepatic steatosis, congestion or cholestasis are easily determined by standard ultrasound evaluation. Currently, inflammation remains the only factor assessable by liver biopsy alone. That's the main reason ultrasound manufacturers have tried to develop such techniques that can assess liver inflammation by using viscosity as a surrogate marker. As we mentioned in the introduction and in the discussion, the clinical utility of determining the viscosity may be in acute alcoholic liver diseases, in differentiating NASH from simple steatosis, detection of allograft damage after liver transplantation. Of course, further studies, referring to liver biopsy, are needed to confirm this.
Example: When dealing with a patient with acute alcoholic hepatitis, due to acute inflammation liver stiffness values of cirrhosis (F4) on elastography can be obtained, even when we have no fibrosis. The patient can be misdiagnosed because the increased stiffness is due to liver inflammation. Therefore, it is vital to understand what causes increased stiffness, fibrosis, or inflammation.
Changes were made in the manuscript (introduction part, page 2, lines 59-70) to highlight the importance of viscosity (necro-inflammation).
- The authors exclude subjects with fatty liver using B-mode ultrasound. Were the subjects with BMI >25 really without any fatty liver? CAP could be got by Fibroscan Compact 530 device. Why did the authors not use CAP to evaluate steatosis?
Although we initially considered standard ultrasound sufficient for selection, EASL recommends conventional ultrasound as a first-line tool for diagnosing steatosis in clinical practice (LoE 1, strong recommendation) and CAP cannot yet be recommended as a first-line technique (LoE 2). But we decided to use the values obtained by CAP combined with standard ultrasound to exclude patients with steatosis. We believe that the changes made add significant value to the manuscript. Thank you for your suggestion.
- Metabolic syndrome or DM, hypertension was supposed to be considered in the exclusion criteria.
Steatosis is considered a hepatic manifestation of metabolic syndrome. We ruled out steatosis using standard ultrasound and CAP. We also excluded other confounding factors such as food intake, cholestasis, venous liver congestion due to cardiac disease, and steatosis. We also decided to change the term used from “healthy subject” to “subjects with healthy livers” to avoid any misunderstanding.
- The relationship between steatosis and liver viscosity is unknown. The authors should address this issue.
This study aimed to determine viscosity values in individuals without known liver disease and without steatosis. Indeed, this aspect is exciting and important, but to do this we need a mixed group that includes patients with NAFLD (in this study all subjects with steatosis were excluded by using CAP and standard ultrasound).
- The normal references of AST, ALT, GGT, cholesterol, triglyceride and platelet should be listed in the manuscript.
Thank you for the suggestion, we made the requested changes.
Reviewer 2
Overall Excellent article and presented in a very nice way. I only have some minor questions and minor edits but overall, I am in the favor of publishing this article.
Thanks for appreciating the article, we tried to edit the article according to your recommendations
Line 72: The structure of statement did not make sense. Do you want to say, “to analyze the effects of gender,age,…” ? In other words, the word ‘and’ needs to be removed.
Thank you, we corrected the mistake.
In Line 89, How did you define alcohol abuse?
We added the criteria used to define harmful alcohol use in the “material and method” section.
In line 288, there is a ‘period’ after the word al which can be removed.
Thank you, we corrected the typo.
In line 290, ‘including the liver’ and not ‘including of the liver.’
Thank you, we corrected the mistake.

Reviewer 2 Report
Overall Excellent article and presented in a very nice way. I only have some minor questions and minor edits but overall, I am in the favor of publishing this article.
Line 72: The structure of statement did not make sense. Do you want to say, “to analyze the effects of gender,age,…” ? In other words, the word ‘and’ needs to be removed.
In Line 89, How did you define alcohol abuse?
In line 288, there is a ‘period’ after the word al which can be removed.
In line 290, ‘including the liver’ and not ‘including of the liver.’
Author Response
We want to thank the reviewers for the kind words of appreciation. It means a lot to us. Also, we want to thank you for the suggestions that have certainly helped us to improve this article significantly. We have made significant changes and hope that the publication of this article will encourage further research on this topic.
Overall Excellent article and presented in a very nice way. I only have some minor questions and minor edits but overall, I am in the favor of publishing this article.
Thanks for appreciating the article, we tried to edit the article according to your recommendations
Line 72: The structure of statement did not make sense. Do you want to say, “to analyze the effects of gender,age,…” ? In other words, the word ‘and’ needs to be removed.
Thank you, we corrected the mistake.
In Line 89, How did you define alcohol abuse?
We added the criteria used to define harmful alcohol use in the “material and method” section.
In line 288, there is a ‘period’ after the word al which can be removed.
Thank you, we corrected the typo.
In line 290, ‘including the liver’ and not ‘including of the liver.’
Thank you, we corrected the mistake.
Round 2
Reviewer 1 Report
The authors have revised the manuscript appropriately according to reviewers' comments.
Minor comments
1. As shown in Figure 2, patient numbers in subgroups can be shown in Figure 4 and 5, as well as the Vi PLUS values, so that the figures can be understood clearly by readers. One or two short sentences to elucidate the concepts in Figure 4 and 5 are suggested in figure legends.
2. The words in line 485 of page 7 can be followed immediately by words in line 486.
Author Response
We want to thank you for your patience and support. We have made the last suggested changes. We hope that in this corrected version the manuscript will be suitable for publication.